# EEG Channel Selection Techniques in Motor Imagery Applications: A Review and New Perspectives

**DOI:** 10.3390/bioengineering9120726

**Published:** 2022-11-24

**Authors:** Ibrahima Faye, Md Rafiqul Islam

**Affiliations:** 1Centre for Intelligent Signal and Imaging Research (CISIR), Universiti Teknologi PETRONAS, Bandar Seri Iskandar 32610, Malaysia; 2Fundamental and Applied Sciences Department, Universiti Teknologi PETRONAS, Seri Iskandar 32610, Malaysia; 3Data Science Institute (DSI), University of Technology Sydney (UTS), Ultimo, Sydney, NSW 2007, Australia

**Keywords:** channel selection algorithm, motor imagery, BCI, electroencephalogram (EEG), biomedical engineering

## Abstract

Communication, neuro-prosthetics, and environmental control are just a few applications for disabled persons who use robots and manipulators that use brain-computer interface (BCI) systems. The brain’s motor imagery (MI) signal is an essential input for a brain-related task in BCI applications. Due to their noninvasive, portability, and cost-effectiveness, electroencephalography (EEG) signals are the most widely used input in BCI systems. The EEG data are often collected from more than 100 different locations in the brain; channel selection techniques are critical for selecting the optimum channels for a given application. However, when analyzing EEG data, the principal purpose of channel selection is to reduce computational complexity, improve classification accuracy by avoiding overfitting, and reduce setup time. Several channel selection assessment algorithms, both with and without classification-based methods, extracted appropriate channel subsets using defined criteria. Therefore, based on the exhaustive analysis of the EEG channel selection, this manuscript analyses several existing studies to reduce the number of noisy channels and improve system performance. We review several existing works to find the most promising MI-based EEG channel selection algorithms and associated classification methodologies on various datasets. Moreover, we focus on channel selection methods that choose fewer channels with great precision. Finally, our main finding is that a smaller channel set, typically 10–30% of total channels, provided excellent performance compared to other existing studies.

## 1. Introduction

Electroencephalography (EEG) in BCI applications is often used compared to other modes such as functional magnetic resonance imaging (fMRI), functional near-infrared spectroscopy (fNIRS), and its low cost and quick response time [1]. Most BCI signals work well from certain places on the scalp. On the contrary, noisy and redundant signals can degrade the BCI efficiency [2,3,4,5,6]. Furthermore, using high channels requires a lengthy preparation time, impacting BCI’s usability. Consequently, picking the fewest channel numbers while maintaining maximum or required accuracy will help to achieve both efficiency and ease. However, identifying the appropriate channel selection is not easy, as manually selecting channels based on neuroscientific data does not always give optimal results compared to using all EEG channels [2].

A brain-computer interface (BCI) aims to convert basic cerebral information into motor commands that a neuroprosthetic device can replicate. Kinesthetic and visual motor imagery (KMI and VMI) are the two basic categories of motor imagery [7]. KMI is characterized by the ability to simulate a movement by creating an impression of how muscles contract and feel during a actual movement. On the other hand, the capacity to see oneself doing the movement is known as VMI. KMI could be more efficient than VMI since most decoders for motor prediction are created based on actual movement. Therefore, employing a KMI to enhance the BCI performance would be preferable. However, individuals could become confused about the visual techniques without explicit instructions. Furthermore, people with motor disabilities had more difficulty using kinesthetic imagery than healthy people [8]. We could categorize the various visual kinds and provide the subject with feedback. With the motor imagery (MI) EEG data, many EEG channel selection algorithms have been developed since 2010 [9,10]. The study of movement and imagery-related activity revealed broadly dispersed frontoparietal cortical regions, the medical aspect of the superior frontal gyrus, the the anterior cingulate cortex, frontal and temporal opercular areas, occipital areas, and the posterolateral cerebellum. In our earlier trial with the sequential movement and imagery (SMI) task, this activity pattern was identical to the usual movement and image activity [11]. As a result, frequency-specific variations in a continuous EEG are event-related phenomena [12].

### 1.1. Context of the Study

The EEG oscillations are classified into bands based on their frequency range. In the context of MI-based BCI, the μ (9–13 Hz), α (8–12 Hz), β (13–30 Hz), and γ (>30 Hz) frequency ranges are the most important. The μ rhythm is a type of α rhythm that exists in the central brain regions, and its morphology is arch-like. Mental imagination of movements, referred to as motor imagery (MI), manifests itself as a result of the rehearsal of a given motor act in the working memory without any overt movement of the corresponding muscle. It is classified into two categories: visual imagery (VI) and kinesthetic imagery (KI). While VI consists of visualization of the subject moving a limb that does not require special training or sense of muscles, KI is the feeling of muscle movement that athletes or specially trained people can generally achieve [13]. When comparing different MI tasks, such as right-hand vs. foot, the μ and β event-related desynchronization (ERD) in cortical areas are identified as the attended body part, and μ and β event-related synchronization (ERS) in cortical areas are identified the non-attended body part. The β, ERS over cortical regions of the treated part of the body is frequently documented [9].

However, its structure and activity are the most specific issues in neuroscience [14]. The BCI faces many challenges, such as computational costs, equipment costs, and classification accuracy [15,16]. Researchers have proposed various techniques to handle these difficulties successfully. Signal preprocessing, feature extraction, and selecting an appropriate classifier, for example, can increase a BCI’s classification accuracy even though multi-channel EEG has a wide range of applications [17], specific low computation complexity, and wearable applications. When building a real-time system, researchers frequently overlook the channel selection phase. The lack of use of a channel selection method means noisy data and redundant channels raise computational and equipment costs for a BCI system. The classification results are also improved or stabilized by channel selection [18]. A similar issue arises when performing feature extraction. As a result, rather than processing and classifying data through all channels, it is critical to choose a practical solution to select the optimal number of channels. After implementing channel selection algorithms, several researchers employed feature selection methods to boost system performance even more [19]. Subset channels are chosen based on criteria that usually include all channel features, such as location, dependencies, and redundancy [20].

### 1.2. Related Works

Researchers have proposed various ways to overcome the problem of reliably selecting EEG channels from EEG data [10]. The employment of the cross correlation-based discriminant criteria (XCDC) algorithm in conjunction with the convolutional neural network (CNN) classifier is one of the most modern ways of selecting EEG channels [21]. The EEG channel selection algorithm (e.g., XCDC, Neuroevolutionary approach, automatic channel selection, and squeeze and excitation blocks (ACS-SE)) is a solid baseline when combined with deep neural networks (e.g., CNN, multi-layer perceptron neural network (MLP-NN), etc.) classifier [21,22,23]. When comparing many EEG channel selection algorithms with various classifiers, it was discovered that DNN and support vector machine (SVM) classifiers produce the best results. As a result, recent contributions have focused on developing ensemble approaches that outperform various EEG channel selection algorithms combined with DNNs and SVM [21,22,24,25]. These methods employ a combination of spatial filters, correlation-based, sequential-based, and binary harmony search-based EEG channel selection, and various classifiers (e.g., CNN [24], MLP-NN [22], SVM [19], linear discriminant analysis (LDA) [26] on one or more BCI competition datasets. A CNN classifier is employed to predict the new hierarchical structure accurately [24,27]. DNN is now widely regarded as the most advanced algorithm for the BCI competition dataset, particularly CNN [21]. The achievement of deep learning is described in numerous classification problems, which encouraged the recent use of deep learning models for EEG channel selection after establishing the present state-of-the-art deep and non-deep learning classifiers for EEG channel selection [24,28].

The experimental paradigm can include both stimulus-dependent and stimulus-independent approaches. MI goes through an internal stimulus-dependent process that creates action representations from information previously stored in long-term memory [29]. In particular, action observation and motor imagery are comparable in forming action representations required to accomplish a skill. However, action observation requires an external stimulus, such as a visual picture, whereas motor imagery does not. Stimulus-independent thoughts (SITs) are streams of ideas and pictures detached from current sensory input. The suggested stimulus-independent hybrid BCI, which combines generated brain signals from the motor and somatosensory cortex, has demonstrated increased performance in individual modalities. The stimulus-independent approach involves more complex probabilistic metrics that detect target EEG patterns [30]. It is worth emphasizing that EEG signals for clinical application rely on practical algorithms for predicting disease or abnormal physiological conditions [31]. As a result, the channel selection section is critical to developing efficient algorithms. Using a small range of channels decreases computational complexity and cost, resulting in low-power devices. We investigated the recently established channel selection algorithms approach for MI-based EEG signals, given the relevance of the channel selection process in BCI systems. Flowcharts and tables support the explanation and conversation, giving the reader a clear comprehension. The classification algorithm, channel selection strategy, and dataset provide a clear and meaningful comparison of several channel selection strategies. Furthermore, the developed method’s performance is described in the classification performance and the number of channels used for aid judgement.

### 1.3. Motivation and Contribution

In this context, the following open questions are addressed in this paper: Which EEG channel selection classifier has the best performance? Is there any strategy for selecting EEG channels that produce state-of-the-art MI data? Which EEG channel selection methods are most effective for the MI task? What effect does the channel selection have on the accuracy? Given that the EEG channel selection community no later addresses questions, it is surprising how a few new papers have overlooked the possibility of resolving EEG channel selection problems with pure learning algorithms [21,24,27,32]. A recent empirical study tested various EEG channel selection techniques on several MI datasets, none of which were deep learning models [10]. This shows how much information on the current performance of deep learning models for tackling EEG channel selection challenges is lacking in the community [24]. In this study, we examine the impact of the EEG channel’s selection algorithm and deep and non-deep learning classifiers on several MI datasets. This study used a 39 EEG channel selection strategy with 19 classifiers in 24 different MI datasets. The following are the main contributions of this paper:We explain that channel selection approaches are specifically designed for detailed EEG channel selection in MI datasets.We propose a unified taxonomy to combine the most recent MI dataset classification applications for multiple EEG channel selection techniques with maximum accuracy.We provide the channel selection direction for the BCI-based paradigm for future research.

## 2. Basic Concepts

We will start with some definitions in this section to make things easier to comprehend. Then, we review the substantial theoretical basis of EEG channel selection for the MI problem. Finally, we show how we used our proposed taxonomy to classify the various channel selection algorithms in various MI datasets. In Figure 1, we provide some guidelines for review methodology for sample collection and analysis.

### 2.1. Basics of EEG Signal

According to studies, the user’s MI stimulates the sensory-motor cortex in the brain, increasing metabolism and blood flow while decreasing or blocking the amplitude of μ (8–13 Hz) and β (14–30 Hz) rhythm EEG signals and oscillation frequency. This is referred to as ERD. On another side of the phenomenon called “event-related synchronization” (ERS), the amplitude of μ rhythm and the β rhythm of EEG signal increases.

Subjective consciousness elicits MI-based EEG, an endogenous evoked activity [33]. It depicts the dynamic process of emotional thought from conception to completion. Similarly, related research in sports rehabilitation suggests that MI training can aid in nerve healing and regeneration of motor nerve routes. As a result, investigating MI-based EEG signals’ processing and use are critical. The difficulty is that EEG and physiological phenomena are more complex than absolute motion, making them harder to detect and treat.

### 2.2. EEG Pre-Processing and Artifact Removal Techniques

Two crucial phases in EEG signal analysis are EEG pre-processing and feature extraction. Pre-processing techniques aid in the removal of undesirable artifacts from the EEG signal, improving the signal-to-noise ratio. By isolating the noise from the actual signal, a pre-processing block aids in increasing the system’s performance. Following that, a feature extraction block aids in retrieving the signal’s most essential features. These characteristics will help the decision-making mechanism get the intended result.

The electroencephalogram (EEG) helps detect brain activity and behavior. However, artifacts in the recorded electrical activity will always impact the processing of the EEG data. As a result, developing ways to recognize and extract clean EEG data during encephalogram recordings is critical. Several approaches for removing artifacts have been proposed. However, artifact removal research remains a work in progress [34].

The paper is organized as follows: Section 2 gives the background of EEG channel techniques. Section 3 defines the different channel selection methods for motor imagery datasets. Section 4 gives a brief introduction to datasets. In Section 5, we discussed the EEG channel selection techniques in detail. We discuss which techniques are good for different MI datasets Section 6. Finally, the conclusion is presented in Section 7.

The present literature review is drawn from well-known databases such as Web of Science (Clarivate) and Scopus (Elsevier). Keywords and queries from credible search engines and databases were assigned to the downloaded articles. The following keywords, for example, were used: EEG channel selection in motor imaging is followed by particular factors to highlight, such as datasets from BCI contests.

Each analysis collected was carefully chosen for this study’s benefit and analyzed to contribute to the literature in this field. The majority of the articles published between 2010 and 2022 were copied. However, specific previously published papers that appear to be worth discussing are included. Articles that did not fall within the scope of the current study were not considered.

## 3. Motor Imagery EEG Signal Channel Selection Techniques

Because EEG signal-gathering equipment is now widely available, BCI based on EEG has become a popular subject for study in recent decades. Since we can capture brain activity data with many channels, EEG equipment is cost-effective compared to other approaches. Researchers can create ways to identify the best channels because there are so many of them. These algorithms are intended to reduce the calculation time, increase classification accuracy, and select the best channels for a specific application or activity. The MI data categorization channel selection task is the subject of this review.

The EEG channel selection methods were taken from feature selection algorithms published in the literature. Feature selection chooses the best subset of features, whereas channel selection employs these methods to choose a channel. After selecting the best channel set, the features are extracted for categorization. On the other hand, the best feature set is fed straight to the classification algorithm in feature selection. The collected features of an optimum channel subset cannot produce good results when selecting a filter channel. To evaluate the credibility of a feature subset, a criterion is used. Since the number of functions is similar to the size of the signal, larger signals have more features. The discovery of an optimal subset is a complicated problem called the non-deterministic polynomial-time (NP) [35].

Our primary purpose is to gain insight related to the use of design methodologies in the development of EEG channel solutions for BCI-based services. Our bottom-up approach for collecting sample articles took a multi-disciplinary perspective. Rather than identifying particular journals, major online databases were selected to source the articles. Figure 1 illustrates the entire sample collection and analysis process.

As illustrated in Figure 2, most feature selection methods have two steps. A heuristic search strategy such as full search, random search, or sequential search is employed to select assessment features during the sub-set production stage correctly. Each new subset of applicant features is reviewed, and its results are compared to the first-best, depending on the classification accuracy.

If the newly selected features outperform the prior one, the new contender will be promoted to the position of the previously selected feature. The process of producing and assessing features is continued until a stop criterion is fulfilled.

### 3.1. Common Spatial Pattern (CSP) Based Algorithm

In this section, we give a brief introduction to CSP methods.

#### 3.1.1. Common Spatial Pattern (CSP)

The classical CSP diagonalizes the two covariance matrices concurrently [36]. Let $X\in R^{M\times N}$ denote a matrix of EEG signals that have been observed, where the channel number is M and the samples are denoted by N. The classic CSP problem is stated as follows:(1)$$\max_{w\in R^{M}}=\frac{w^{T}c_{1}w}{w^{T}c_{2}w},$$
where *w* is a spatial filter coefficient, $C_{i}\left(i=1,2\right)$ indicates a single-class covariance matrix. Generally, the generalised eigenvalue decomposition (EVD) can handle this problem.
(2)$$C_{1}w=\lambda \left(C_{1}+C_{2}\right)w,$$
where λ is an eigenvalue of $C_{1}$ and $C_{2}$. Moreover, *M* eigenvectors are generalizations obtained by solving Equation (Equation 2). In practice, we use the first *r* eigenvectors ${\left\{w_{i}\right\}}_{i=1}^{r}$ and the last eigenvectors *r* and ${\left\{w_{i}\right\}}_{i=M-r+1}^{r}$ as spatial filters, if the eigenvalues ${\left\{\lambda_{i}\right\}}_{i=1}^{M}$. They are organized according to their size in a non-ascending way. We ultimately define $W=[w_{1},\ldots ,w_{r},w_{M-r+1},\ldots ,w_{M}]$. These issues led to the choice to spread the spatial filters of the CSP, focusing on a few channels with significant class variations and avoiding channels with low or irregular noise or artifact variations. The rows of the CSP projection matrix assigns uniform channel weights to maximize discrepancies between two EEG signal types. Therefore, the vectors of the source distribution can be considered the CSP space filters.

#### 3.1.2. Sparse CSP

Due to classic CSP inadequacies, several researchers aim to integrate sparsing behavior with conventional CSP to discover and eliminate highly noisy or interfering channels. Given *w*’s sparsity *k*, i.e., the number of nonzero items in *w*. The sparse CSP problem is stated as follows:(3)$$\max_{w\in R^{M}}\frac{w^{T}c_{1}w}{w^{T}c_{2}w}\;s.t\;{∥w∥}_{0}=k.$$

The ${∥.∥}_{0}$ is the Euclidean distance and the problem can be converted into Equation (Equation 2) typical problems if *k* channels are specified. Where $C_{1,k}$ and $C_{2,k}$ are the *k* × *k* sub-matrices of $C_{1}$ and $C_{2}$. However, this is generally impractical. To tackle such a problem, further develop forward selection (F.S.), reverse elimination (R.E.), and recursive weight removal (RWE) [37].

#### 3.1.3. Regularized CSP

A regularized CSP (RCSP) approach is recommended to regularize the covariance matrix estimate in CSP extraction. Regularized covariance-matrix estimation is employed in the Regularized CSP algorithm in RCSP by applying the regularization technique presented in the general learning concept [38,39]. The CSP algorithm is regulated in a small sample environment (SSS). There are two regularization parameters used in [38]. The first regularization parameter orients the reduction of a specific subject covariance matrix to a more general covariance matrix to lower the estimated variance. This is based on the [39]. The second parameter of regularization limits the reduction of the sample-based covariance estimate in the direction of a scaled identification matrix to the bias due to the restricted number of samples. In addition, the challenge of determining regularization parameters for RCSP must be solved. On the other hand, the number of samples in SSS may not be enough for regularization parameters to be determined by the approach, which is used commonly [38]. Consequently, the tensor object recognition aggregation technology is employed to identify the regularization parameters in the EEG signal classification using RCSP that aggregates various regularized CSPs to generate a solution based on an ensemble [40].

### 3.2. Correlation Based Algorithm

The study helps to choose highly corresponding EEG channels for each patient against one reference channel without affecting classification accuracy. An individual channel sub-set provides a more efficient classification while lowering the computing complexity and time.

#### 3.2.1. Correlation Coefficient

Spectral entropy is a theoretically describable measure of signal disorder: the correlation coefficient of spectral entropy of motor imaging was employed to identify series channel groups. The Filter Bank Common Spatial Pattern (FBCSP) algorithm assessed the performance of the channel groups.
(4)$$H\left(E\right)=-\sum_{i}^{N}log_{10}p\left(E_{i}\right).$$

The probability is $E_{i}$, where $E=E_{1},E_{2},\ldots ,E_{N}$ is the signal in the $P(E_{i})$ time domain. The algorithm calculates the signal of each layer based on the autoregression model. This investigation uses the correlation coefficient approach to picking channels using the “interested class vs. the rest” strategy. EEG is divided into two groups, $s_{1}$ and $s_{2}$, which includes the $s_{1}$ interest group. $H_{1}$ and $H_{2}$ are spectral entropy identified to match $s_{1}$ and $s_{2}$. “Spectropic entropy correlation” is the relationship between the two groups $s_{1}$ and $s_{2}$. This is a measure of how tightly these two groups, represented as
(5)$$\rho \left(H_{1,j},H_{2,j}\right)=\frac{cov(H_{1,j},H_{2,j})}{\sigma_{H_{1,j}}\sigma_{H_{2,j}}},$$
where $\sigma_{H_{1}}$ is the standard deviation of the spectral entropy, and *j* is the index of the channel. We calculate the spectral entropy for each channel in all frequency ranges for selection by employing a total square correlation coefficient.
(6)$$P\left(H_{1,j},H_{2,j}\right)=\sum_{i=1}^{N}\rho^{2}\left(H_{1,j},H_{2,j}\right),$$
where a spectral entropy estimate of i=1,2,…,N is the number of frequency bins. The channels are chosen based on the channels’ $\rho (H_{1,j},H_{2,j})$ correlation coefficient rating. The channels picked for $C_{i}$. It will be added to the FBCSP algorithm.

#### 3.2.2. Pearson’s Correlation Coefficient (PCC)

The Pearson correlation coefficient is a statistical association or linear dependence between two or more random variables [41]. It is defined as follows:(7)$$\rho \left(X,Y\right)=\frac{1}{n-1}\sum_{i=1}^{n}\left(\frac{X_{i}-\bar{X}}{\sigma_{X}}\right)\left(\frac{Y_{i}-\bar{Y}}{\sigma_{Y}}\right).$$
when the two variables are *X* and *Y*, *n* is the number of observations, *X*, and *Y* is the means of both $\sigma_{X}$ and $\sigma_{Y}$. These are the default deviations between the two. In this example, the value of ρ(X,Y) is 0 to 1, which shows that the relationship to the value is low to high. The correlation coefficient is measured for each pair of EEG channels.

#### 3.2.3. Cross-Correlation Based Discriminant Criterion (XCDC)

Signals from the same MI task class should have similar functionalities in the MI EEG classification procedure and vice versa. As a result, we may evaluate a channel’s discriminating performance by comparing signals from different classes. They suggested a signal cross-correlation-based channel selection approach based on this premise. Yu et al. [21] described the details about XCDC.

#### 3.2.4. Canonical Correlation Analysis-Channel Selection (CCA-CS)

The association of multivariate functional groups with target classes can be evaluated by CCA [42]. The CCA focuses on the different MI-based tasks and distinguishes between different movements. The maximum linear correlation, i.e., CCA $\left(S_{P},Y\right)$, is used to determine the connection between the $S_{P}$ channel group and the *Y* goal class vector.

### 3.3. Sequential Based Algorithm

These algorithms examine the functional area in its entirety to identify the top features. The most common strategy was sequential function selection (SFS), and adding the function with the highest value for the target function was initially empty [43]. The additional features and evaluation of the new subset are in the next step. The SBS sequence was the reverse. The SBS reversed SFS, started with all the features, and deleted those that had the most negligible effect on the performance of the target function [44].

#### 3.3.1. Sequential Floating Forward Selection (SFFS)

The sequential forward floating selection (SFFS) was a more flexible approach, adding an update step next to SFS [45]. After deleting one characteristic from the subset, the backtracking phase analyzed the new subset. If the deleted feature maximized the objective function, the algorithm returned to the first stage using the new reduced features. If not, proceedings would continue until the number of features or performances it wishes to achieve. The nesting effect was significant in the SFS and SFFS algorithms. There could be two significantly correlated features because they offer the best precision in the subset. The EEG electrode channels were chosen using sequential floating forward selection (SFFS) [46]. The selection criteria were a combination of optimizing accuracy and minimizing costs. To distinguish between channel combinations, CCA accuracy was utilized. It was a process that was repeated. SFFS could add or delete electrodes from the existing set at each cycle and iterate until the desired number of channels was chosen. When the number of characteristics is excellent, the SFFS is time-consuming. A selection function can be observed in adjacent channels based on the distribution of channels in the cerebral cortex. The whole set cannot select or delete multiple channels from the SFFS improvement at once [46]. The critical distinction between SFFS and improved SFFS approaches is that improved SFFS methods have fewer features. The amount of time it takes to find anything might be drastically decreased.

#### 3.3.2. Generalized Sequential Forward Selection (GSFS)

The GSFS approach is used to identify the optimal channels in this work [47]. For starters, all channels are given their frequency band characteristics and their CSP. Then, in the first phase, two channels with the best performance combined are chosen, using the classifier accuracy as a criterion for channel selection. The channels chosen for the previous step are added to the following channels, and the system’s classification performance is calculated. This is repeated for each channel until the optimal channel combination is found.

#### 3.3.3. Bhattacharyya Bound and Sequential Forward Search (B.B. and SFS)

The Bhattacharyya standard spatial model (CSP) bond is employed to create the ideal index, and a rapid sequential forward search obtains the optimum channel combination [47].

### 3.4. Binary Particle Swarm Optimization Based Algorithm

In 1995, the PSO algorithm was devised by Kennedy and Eberhart [48]. It is based on a bird social comportment simulation. Particles are defined as a possible solution for the search space and are flown throughout a hyperdimensional search room. The particles go across the search space at a certain velocity. The discoveries and prior experience of the other school members may assist individual swarm members. The current and preceding solutions are considered perfect when each portion looks for an optimal solution in the search space. Therefore, the speed of every particle depends on its own best location and its neighbors’ best solution. Each particle searches for the optimum solution and updates using the fitness value. Gbest is the best solution for swarming, and pbest is the most excellent particle option. Finally, the swarm converges to inappropriate places. The position and speed of the particles are updated as follows:(8)$$v_{k+1}^{i}=wv_{k}^{i}+c_{1}r_{1}\left(p_{k}^{i}+x_{k}^{i}\right)+c_{2}r_{2}\left(gbest-x_{k}^{i}\right)$$
(9)$$x_{i}\left(t+1\right)=x_{i}t+v_{i}\left(t+1\right),$$
where *i* refers to every particle. $c_{1}$ is cognitive and $c_{2}$ is a social element. $c_{1},c_{2}$, which checks how far a single test particle is going, is the constant value. *W* is an inertial weight that limits the previous speed. $r_{1},r_{2}$ Random values between 0 and 1 are arbitrary. When $c_{1}>c_{2}$ updates speed weights on the forces from a tendency to return to its best possible solution so far, which is much greater than the force of attraction of the best solution in the neighborhood.

PSO has a discrete version called BPSO [49]. The velocity is updated similarly to PSO. The only difference between PSO and BPSO is that in BPSO, the particles are either 0 or 1, and the update rule for each position is different. The following are the updated equations:(10)$$S\left(v\right)={\left(1+e^{-v}\right)}^{-1}$$
(11)$$x_{k+1}^{i}=1\;if\;\tau <S\left(v_{k+1}^{i}\right)$$
(12)$$x_{k+1}^{i}=0\;if\;\tau >S\left(v_{k+1}^{i}\right),$$
where τ is between 0 and 1, a random value, the channel space is considered the solution space, and the value of each portion can be 0 or 1. PSO must be altered (which will be generically called MOPSO). MOPSO is an unobtrusive binary version, while BMOPSO is an unobtrusive binary MOPSO version. Figure 3 describes the summary of the channel selection algorithm used in this paper.

## 4. Motor Imagery EEG Datasets

Electrodes represent the EEG signal acquisition unit, whether invasive or non-invasive. The datasets and channels used in this empirical investigation are described first in this section. We then go over our open-source MI classification channel selection architecture in depth. There were 23 separate MI-based EEG datasets used in the examined studies, most of which are publicly available, and some are private.

The first to introduce the BCI Competition III Datasets IVa was Fraunhofer FIRST, the Intelligent Data Analysis Group (Klaus-Robert Müller, Benjamin Blankertz, and the Charite-University Medicine Campus Franklin, the Neurology Department of the Neuro-Gabriel Curio Group (Gabriel Curio). BCI Competition III dataset IVa contains EEG signals from five right-hand and right-foot MI subjects. 280 EEG tests were recorded from 118 electrodes placed in comprehensive international 10–20 systems for every subject at a sampling rate of 1 kHz. The experiment was carried out according to a classical paradigm. A visual index of three tasks begins each trial (left hand, right hand, and right foot). Subsections were invited before a random length between 1.75 and 2.25 s to relax for 3.5 s to perform the corresponding MI task.

Dataset 1 of BCI competition IV of seven healthy topics was recorded. Two MI classes, including left hand, right hand, and foot, have been selected for each subject. A set of EEG signals from 59 channels is available for each subject. Each subject’s dataset consists of two parts: data for calibration and analysis. We select calibration data from subjects a, b, d, and e for our experiments to check algorithms containing 200 experiments.

Dataset IIIa BCI Competition III was provided by the Graz University of Technology’s Laboratory of Brain-Computer Interfaces (BCI-Lab) [50]. Over 60 channels with a sampling rate of 250 Hz of three participants, k3, K6, and L1, were recorded in dataset IIIa of BCI Competition III. The first 2 s were used as a pre-stimulation baseline in each trial, which instructed participants to rest. An acoustic stimulus is presented in t = 2 s, with a fixing cross. From t = 3 s, there has been an arrow pointing left, right, up, or down for 1 s, and a left, right hand, tongue, or foot motion was asked to imagine until the cross has disappeared at t = 7 s.

The CLA dataset includes a left/right paradigm for the Classical (CLA) MI tests. EEG signals in the standard international 10–20 system with two ground electrodes (A1 and A2) and a syncing channel that does not contain actual EEG data are recorded at a sampling frequency of 200 Hz from 19 electrodes. A visual signal with the left, right, or passive response is displayed on a screen at the beginning of each test, with a response from MI required by the subject.

Commercial dry and wet electrodes have been utilized to collect EEG data in various BCI applications and investigations [39,40]. Commercial dry electrodes, on the other hand, due to their high contact impedance, cannot be used in several applications that demand a high-quality signal. These datasets differ in terms of the number of electrodes, subjects, and total trials, among other things; Table 1 summarizes the datasets. Figure 4 details the percentage of different datasets used in this article.

## 5. Motor Imagery EEG Classification for Channel Selection

The MI test is crucial for motor injuries in patients. The EEG signals may be used for this kind of analysis. The channel selection may entail determining which channels are most relevant to a specific cognitive activity while decreasing the overall computing complexity of the system.

### 5.1. Methods Based on CSP Variants

Common Spatial Pattern (CSP) filters are frequently employed to examine MI-EEG in the literature. The CSP can maximize the variance of two classes (Koles, Lazar, and Zhou 1990). Chen et al. [51] adjusted the CSP approach to extract the correct number of channels. The approach determined which channels in the 43 frequency bands had the greatest impact on the classification for each subject. Each frequency band’s CSP characteristics were computed, and all frequency bands’ CSPs were connected in series. The channel selection technique is independent of the classification results based on the similarity measurement due to the shorter calculation time and sensitive feedback. The threshold-based channel selection approach revealed that a subset of channels could produce the best results for the CSP and SVM graders. In the experiment, the Dataset IIa BCI Competition IV, Dataset IIIa BCI Competition III, and Dataset IVa BCI Competition III were used, and an SVM classifier was used for classification, with an average accuracy of classification 77.82%, 86.02%, and 86.86% simultaneously [51]. CSP weights to assess whether they could be used for EEG sensor selection compared to regularized procedures. Six electrodes out of 60 are chosen for filter-based function extraction and have attained an average accuracy of 83.70% over LDA [52].

Tam et al. [53] select channels by classifying the CSP or CSP coefficients. For two classes, the CSP generates two filters. The CSP employed these coefficients to produce weights for new filtered signals on other channels. When the weight of a particular electrode was significant, it was supposed that the filtered signal contributed further and that the electrode was, therefore, considered essential. The first electrode was the most outstanding value from the sorted coefficient of the class 1 filter. The second channel was supplied by the sorted coefficients of filter class 2. The process is used if the channel has been selected until a new channel is chosen to reach the next most crucial coefficient in the same class. The data were gathered from five chronic stroke patients during 20 MI work sessions using a 64-channel EEG headset of 250 Hz, each on a particular day [54]. The proposed approach showed an average classification of 90% of Fisher Linear Discriminant (FLD) classification with electrodes ranging from 8 to 36, compared to 64. The best categorization for 22 electrodes was 91.70%. The recommended CSP-R-MF method focused on different changes in MI brain zones in different frequency bands [55]. It used the CSP-rank approach to pick channels for each frequency band automatically. Dataset 1 BCI Competition IV and Dataset IVa BCI Competition III attained an average accuracy of 82.48% and 77.75%, respectively, with the LDA classifier. The regularization of the CSP approach proposed the selection of 24 channels out of 118 with a 93% average accuracy [56].

On Dataset IVa BCI Competition III, a comparison was made with five existing channel selection algorithms to determine the creditability of the proposed methods. The RCSP algorithm could shortlist the 24 channels and achieve a 93% accuracy. RCSP outperformed SCSP by 12.22%, CSP-R-MF by 15.25%, CSP with l1 norm by 3.32%, and CSP by 6.14%, according to the previous discussion. Table 2 shows the summary of the methodologies of CSP channel selection for EEG applications.

The regularization parameter regulates the number of selected channels in the suggested SCSP optimization problem [1]. As a result, changing the regularization value results in a different number of selected channels. The results showed that the suggested SCSP method that used the first criterion achieved the best 80.78% grading accuracy by removing most of the channels and 22 out of 118 channels. The Robust Sparse CSP (RSCSP) solution to resolve BCI channel session selection difficulty was suggested by [56]. Based on previous experience, the pre-selected channel subset was picked. A robust minimal covariance determinant (MCD) estimate that contained an outlier’s resistant parameter was replaced in the SCSP covariance matrix.

**Table 2 bioengineering-09-00726-t002:** Common spatial pattern (CSP)-based techniques for EEG channel selection of motor imagery EEG channel selection.

Techniques	Channel Selection Strategy	Classifier	Accuracy (%)	No. of Selected Channels/Total No. of Channels	Dataset
Meng et al., (2009) [54]	CSP with l1 norm	SVM	89.68	20/118	Dataset IVa BCI Competition III
Tam et al., (2011) [53]	CSP Rank	Fisher Linear Discriminant (FLD)	91.7	22/64	BCI-FES training platform
Arvaneh et al., (2011) [1]	Recursive Fearture Elimination using Sparse CSP (SCSP)	SVM	81.63 (SCSP1), 79.09 (SCSP2) 82.28 (SCSP1), 79.28 (SCSP2)	13/22 9/22 23/118 8/118	Dataset IIa BCI competition IV Dataset IVa BCI Competition III
Arvaneh et al., (2012) [56]	Recursive Fearture Elimination using Robust Saparse CSP (RSCSP)	SVM	70.47	8/27	Stroke patients EEG dataset
Saha et al., (2016) [57]	RCSP	WC Classifier	93	24/118	Dataset IVa BCI Competition III
Masood et al., (2017) [52]	CSP Weights	LDA	83.70	6/60	Dataset IIIa BCI Competition III
Feng et al., (2019) [55]	CSP-R-MF	LDA	82.48 77.75	24/59 30/118	Dataset 1 BCI Competition IV Dataset IVa BCI Competition III
Chen et al., (2020) [51]	CSP	SVM	77.82 86.02 86.86	15/22 24/60 30/118	Dataset IIa BCI Competition IV Dataset IIIa BCI Competition III Dataset IVa BCI Competition III

### 5.2. Correlation-Based Techniques

The Pearson Correlation Coefficient (PCC) approach was introduced, which calculates the relationship of EEG signals to highly correlated EEG channels for a specific patient with no sacrificing accuracy in classification [58]. A total of 280 studies were conducted on each of these five participants, with an EEG of 118. A reference for extracting characteristics for one of the three channels, C3, C4, or Cz, is utilized, and its correlation with the other channels is calculated. It is worth noting that the correlation of 0.7 was discovered following a preliminary analysis that used C3, C4, or Cz as a reference channel result. The average accuracy for Dataset IVa BCI Competition III was 74.52% using the LDA classifier with 30 out of 118 channels and 84.01% using 39 out of 60 channels for Dataset IIIa BCI Competition III.

Jin et al. [59] describes a correlation-based (CCS) approach for selecting channels with more associated information. The objective is to improve the performance of the BCI classification based on MI. In addition to extracting valuable characteristics, a unique regularized common spatial pattern (RCSP) technique is applied. The contribution of this study is to propose a filter based on an analysis of correlations to reduce irrelevant channels and extract information via RCSP from selected channels. Conduct testing to test the successful use of the given methodologies through training the Radial Basis Function (RBF) kernel support vector machine (SVM) classifier. This approach (Jin et al., 2019) achieved an average accuracy of 81.60% for Dataset 1 BCI Competition IV with 30 out of 59 channels, 87.40% for Dataset IVa BCI Competition III with 42 out of 118 channels, and 91.90% for Dataset IIIa BCI Competition III with 19 out of 60 channels.

The new channel selection approach, XCDC for MI EEG classification [21]. The technique suggested selecting the most discriminating channels based on the cross-relation of signals from repeated EEG tests. XCDC is quadratic in complexity. To obtain a discriminating score of the channel, the correlation between every pair of studies must be calculated to result in quadratic complexity. Only the specified channels, decreased number of channels, configuration complexity, and computational costs will be necessary for future sessions or applications. Wang et al. [42] submitted an SVM-CCA-CS algorithm and examined the optimum CS on the motor screen for multi-channel EEG signals. The initial extraction of the Wavelet Packet Coefficients and features, the weights of each feature group were determined, and CCA-CS predicted the starting weight of each channel. The original channel weights were subsequently changed based on the current accuracy of the SVM classification. The results show that our proposed method can select the most ways to achieve classification accuracy, indicating that EEG signals from some ideal ways can also ensure classification accuracy. The initial 30 channels achieve an average accuracy of 80.03%, accounting for only 21.20% of the 118 acquisition channels. Yang et al. [43] applied a correlation coefficient channel selection approach ranking to determine the ideal channel combination and enhance classification performance by 1.25% to 8.22%. Using BCI technology, both healthy and ALS individuals increase their skills. The analysis shows that positioning the selected channels ensures that the participants coincide with the location of one or both motor cortices. These people may perform MI tasks by engaging their motor cortices consistently throughout tests. The summary of correlation-based channel selection techniques for MI-based EEG applications is shown in Table 3.

### 5.3. Sequential Based Techniques

Pudil et al. [44] used sequential floating forward selection (SFFS) to choose the EEG electrode channels. The selection criteria were a combination of optimizing accuracy and minimizing costs. Distinguishing between channel combinations does not modify the subject’s accuracy and repeats the process. SFFS could add or delete electrodes from the existing set at each cycle. SFFS also iterated until the desired number of channels was chosen.

SFFS deletes the most useless characteristic dynamically from the chosen function subset and inserts the most significant characteristic from other functionality into the chosen feature subset. The SFFS can therefore be selected for the channel. However, this is a complex strategy, though, especially with many features.

Reunanen et al. [45] offers the modified SFFS to pick the ideal channels to overcome this issue. The most senseless functionality is dynamically eliminated from the specified sub-set of features, and the most significant feature is inserted from the rest into the selected feature subset. The SFFS is hence suited for selecting the channel. This method is quite demanding, though, mainly if there are many features.

Radman et al. [61] chose the optimum channels using generalized sequential forward selection (GSFS). To begin, all channels’ frequency bands, and CSP characteristics are determined. Then, in the first phase, two channels with the best performance combined are chosen, using the classifier accuracy as a criterion for channel selection. The channels chosen for the previous step are added to the following channels, and the system’s classification performance is calculated. For each channel, this is performed until the best channel combination is identified. The GSFS technique improves its mean accuracy compared to the case in which all channels are employed.

He et al. [47] introduced a Bhattacharyya bound-based sequential forward-searching technique. The Bhattacharyya border of a common spatial pattern is the optimal index, and a quick, sequential forward look is used to identify the ideal channel combination. This approach requires the preliminary treatment of EEG using a standard average common average reference (CAR) to remove artifacts and noise, followed by portability filtering at 7 to 30 Hz frequencies (containing mu and beta rhythms) and employing 3-fold cross-validation for comparison with Bhattacharyya. A naive Bayes (N.B.) classifier labels the CSP features retrieved from the channels chosen using this technique in each fold. The CSP feature vector has a dimension of 6. The accuracy of the classification curves is three times based on Bayes 3-fold cross-validation, each of which consists of randomly selected specimens. Repeat the cross-validation approach ten times for each participant. Table 4 summarizes sequential channel selection approaches for EEG engine imaging applications.

### 5.4. Particle Swarm Optimization Based Techniques

An MI-based brain/computer interface is a device that classifies EEG signals gained out of the imagination of limb movement to transform a person’s intention into a control signal. We do not know which positions are engaged or not in the new paradigm. Using as many channels as possible is a straightforward solution. Other issues arise because of the use of many channels. Many channels pose an over-fitting problem when utilizing a common spatial pattern (CSP)—an EEG extraction approach. This technique for medical analysis is very challenging to use. We used particle swarm optimization on CSP [63] to solve these issues. 7 healthy people participated in the experiment, with 59 electrodes, and dataset 1 BCI dataset IV was utilized. Two were chosen from the three MI classes for each subject (left hand, right hand, foot). The accuracy of each subject was measured five times for each channel selected by BPSO, and 52% improved with the recommended strategy for optimal channel selection overall channels. The accuracy was 20% more excellent with 30 selected channels than all channels.

Multi-channel EEG signals require a complex and unpleasant recording process as the number of channels increases and affects classification accuracy. To solve this issue, we introduced a unique approach to channel reduction called binary multi-objective particle swarm optimization (BMOPSO), which optimizes the number of selected channels and mutual information [64]. Cross-validation of 10 × 5 has been used to obtain the classification precision rate for classification performance assessment. The k-NN method has an accuracy rate of 83.40%, slightly higher than the SVM algorithm, and selects eight channels from 22. Table 5 shows the summary of strategies for PSO-based channel selection for MI-based EEG applications.

### 5.5. Other Methods

Shapelet-transformed EEG Channel Selection (StEEGCS) was introduced to select EEG channels [25]. EEG is a brief sequence representing the original EEG data and divides EEG into groups according to its distance from the EEG forms. Shapelet is a time series subsequence often used in series-time data extraction [25,65,66,67]. On the other hand, the EEG form is a continuous EEG subsequence that is much shorter and inherits the structures of the original EEG. They found, in comparison to the baselines of 10 EEG datasets, that three EEG channels, the three with a length of 30, can select the most appropriate EEG and yield the most appropriate classification of EEG channels and the better classification performance of the EEG channels selected so that they set the best EEG channels. The optimization problem by jointly learning the optimal EEG shapelets *S*, channel contributions π, and the optimal hyperplane *W*, simultaneously,
$$\min_{S,\pi ,W}F=\sum_{i=1}^{N}\sum_{v=1}^{V}+\frac{\lambda_{W}}{2}{∥W∥}^{2}+\frac{\lambda_{S}}{2}{∥A∥}^{2}.$$

Examine the impact of the StEEGCS-selected channels on EEG classification accuracy using SVM first. The results show that the accuracy of the sampled EEG channels on all EEG datasets increases typically with three shapelets of a length of 30. The channels selected by StEEGCS further reveal that the accuracy of EEG grading for every EEG dataset improves by 9.5% compared to non-selective channels (i.e., all channels). StEEGCS aims to locate separate EEG forms, which are the original EEG data, to give the SVM classification more usable EEG patterns. In other words, when shapelet-transformed StEEGCS is integrated with logistic loss, it eliminates duplication in EEG data and strengthens significant patterns for classifier modeling. In the meantime, the performance of the SVM classifier improves compared to non-selected EEG channels as the number of picked EEG channels decreases. The computational complexity of StEEGCS is $O(I_{iter}\left(nckc^{2}l^{3}+nvck^{3}l^{3}\right))$, which indicates a significant number of iterations(or time) to identify the ideal duration and number of shapelets. The time consumption of StEEGCS is predominantly driven by optimum format learning. *k* denotes the number of shapelets that need to learn; *v* denotes the number of EEG classes; *c* denotes the number of EEG channels; *l* is the maximum length of shapelets. The study of optimal channel selection and the neural network weight end-to-end through the Gumbel–SoftMax technique on the EEG channel selector layer [24,68]. In conjunction with a baseline EEG selection mechanism tailored to this job, the Gumbel-SoftMax method is proposed: mutual information and gullible forward selection with usefulness measurement. He proposed to use a concrete selector layer to resolve the EEG channel selection problem from end to end [69]. This method incorporates channel selection into the model’s training process, addressing its discrete character with categorical reparameterization. Two different EEG tasks were used for the method’s performance: motor performance and auditory matching.

A Cohen’s d effect size CSP (E-CSP) based channel selection approach was proposed [70]. The approach filters out channels that do not provide any meaningful information. First, before determining the effect size of Cohen’s d channel selection, it decreases the noisy trials for each channel. The noisy tests were removed via the z-score approach, which computed the distance between each test and the middle. The test was far from the mean and might be labeled as noisy. Cross-validation was used to determine the z-score cutoff. For choosing channels, Cohen’s d effect size was used. The distance of Cohen’s d was established using the following formula:(13)$$d_{i}=\frac{\left|\right|\bar{C_{1i}}-\bar{C_{2i}}\left|\right|}{\sigma }$$
where,
(14)$$\sigma =\frac{\left(\sigma_{1i}^{j}-\sigma_{2i}^{j}\right)}{2}.$$

Across the selected *j*th trials, the standard deviation of 1 class of channel *i* is represented by *j*th. ${\bar{C}}_{ii}$ is the *l* class average of *i*th. If the *d* value was more significant than δ, the channel was chosen. $L=\{i\;:d_{i}>\delta \};\forall i$ denotes the set of channels to be used. The suggested range for δ is [0.01–0.1], calculated using cross-validation. To extract features, CSP was implemented. To test the algorithm, two datasets were used. The first is from competition BCI IV, Datasets IIa, and the third is from BCI III, IVa Dataset [71,72]. The SVM classification outcomes were compared to the CSP techniques of SCSP1 and SCSP2. The results showed that the algorithm recommended exceeded 3% of other algorithms for a dataset I on average, with an accuracy of 83.61% on eight channels. For dataset 2, the average classification precision with 9.20 channels was 85.85%. The results demonstrated that, compared with earlier procedures, the strategy enhanced classification accuracy while lowering selected channels.

Parshiva et al. [73] describe an approach to decreasing EEG dimension for MI-BCI by training each participant to autonomously encode EEG data optimally via an artificial neural network with a sub-complete, sparse autonomous autonomic encoder. A smaller fraction (t = 0.5 s and t = 3.5 s) of the MI-based task data for each trial is further evaluated. The average rating accuracy of only 13 encoded channels for ten patients is 74.3 ± 8.06%. Following cross-validation, we see that the independent autocoder has an average ranking precision of 66.64 ± 3.93% with just 11 channels. The research expands automatic encoder-neural networks in the motorized brain-computer interfaces and demonstrates that performance increases dramatically when data dimensions are dropped. The changes in mental tasks (imagination of the left and the right) vs. relaxation times are measured by Itakura distance (I.D.) [74]. The EEG data were acquired in conjunction with the BCI2000 platform utilizing the g.MOBIlab+ module from G.Tech Guger Technologies [75]. After statistical tests are performed, channels are selected. The linear discriminant analysis (LDA), the quadratic discriminant analysis (QDA), and the Mahalanobis Distance (MD) Classifiers are used to categorize the data on the specific channels. The QDA classification provides the highest accuracy for classification. Table 4 presents an overview of additional MI-based EEG channel selection strategies.

## 6. Discussion and Guidelines

Selection algorithms and procedures. This article examined MI-EEG methods for selecting channels with various strategies that considered various parameters for channel evaluation and search strategies stated in the literature. This review paper provides the advantages and disadvantages of the various methodologies for channel selection employed in MI-based EEG applications. The extensively investigated channel selection methods show that a limited selection of EEG channels with a small quantity of pre-calculation can be used to attain decent performance metrics. According to the study, the use of a channel selection technique can reduce the number of channels by up to 80% without significantly impacting categorization jobs. Reducing the number of channels will minimize computer complexity and reduce setup time. It enhances the maintenance of the gadget for the subject well.

The most well-known MI EEG datasets are dataset IVa BCI competition III and dataset I BCI competition IV; most of the experiments examined employed these two datasets. Other MI EEG datasets are also accessible, however less frequently used, such as datasets IIIa BCI competitive III, datasets IIa BCI competitive IV, High gamma, low limb motion, and patients with Amyotrophic Lateral Sclerosis (ALS). Electrical signals on the scalp vary according to physiological tasks. One of these distinctions is the frequency range of the signals associated with specific tasks. Different frequency ranges for dealing with MI tasks during the analysis process have been suggested in studies. The range was observed to be within 0.05–200 Hz in the analyzed research, but several studies were conducted at high frequencies, e.g., the high-gamma dataset. Table 2, Table 3, Table 4, Table 5 and Table 6 summarize the strategies used for MI-based EEG applications in channel selection, as discussed in Section 5.

Several databases have been used to test these strategies. An extensive investigation will be required to determine the efficiency of an algorithm in all MI-based EEG applications. Finally, it was discovered that 12 of the 38 strategies used for channel selection attained an average accuracy rate of better than 90%. Yu et al. [21] has shown the excellent channel selection approach cross-correlation-based discriminant criterion (XCDS) with classifier convolutional neural network (CNN) to achieve the most excellent accuracy of 99.64% and 99.28% on dataset IVa BCI competition III and (CLA) left/right hand MI dataset. This approach achieves a channel selection rate of more than 50%. This method’s advantage is that the EEG channels are quantified and classified in MI tasks as necessary, providing a practical method to select the classification channels of MI BCI systems during the calibration phases, thus, alleviating computational complexity and configuration difficulty during subsequent steps, leading to more convenient, real-time BCI systems. It is impossible to find the optimal subset of channels by assessing any combination. These methods provide a sorted channel list sequentially or according to a number criterion. This method offers a helpful way of resolving the two problems: specify reference accuracy and tolerance ratings in the calibration phase and select the subset with the least number of channels that satisfy the requirements. In this method, we use all-channel precision to evaluate the tolerance rates of 5%, 1%, and 0%. However, these two parameters could be arbitrarily identified according to current application requirements. This method will degenerate to select a setup with maximum accuracy when the highest accuracy is chosen, and the tolerance rate is set to 0. The disadvantage of this method, despite the encouraging performance and efficiency, is how to compare channel ranking methods and choose a subset given the results. The two central aspects of analysis are statistical, and machine learning/deep learning analysis upon different datasets as follows:

**Table 6 bioengineering-09-00726-t006:** Other techniques for motor imagery EEG channel selection.

Techniques	Channel Selection Strategy	Classifier	Accuracy (%)	No. of Selected Channels/Total No. of Channels	Dataset
He et al., (2013) [23]	Rayleigh coefficient maximization Based Genetic Algorithm	Fisher Discriminant Analysis (FDA)	88.2 89.38	29/118 12/32	Dataset IVa BCI Competition III MI experiments
Shenoy et al., (2014) [19]	Min. Redundancy Max. Relevancy (mRMR)	SVM	90.77 81.22	10/118 10/22	Dataset IVa BCI Competition III Dataset IIa BCI Competition IV
Das et al., (2015) [70]	Cohens d effect size	SVM	85.85	9/118	Dataset IVa BCI Competition III
Eva et al., (2015) [74]	Itakura Distance Method	Quadratic Discriminant Analysis (QDA)	96.6	8/20	G.Tech Gugar Technologies Together with BCI2000
Liu et al., (2017) [26]	Fisher’s criterion	LDA	93.50	5/33	MI dataset for patient with Amyotrophic lateral scelerosis
Joadder et al., (2018) [76]	Wavelet Energy	LDA SVM K-NN	75.6 76 69.9	3/118	Dataset IVa BCI Competition III
Gurve et al., (2019) [77]	Non-Negative Matrix Factorization (NMF)	LDA	96.66	13/19	Lower limb MI dataset
Parshiva et al., (2019) [73]	Auto-encoder	Auto-encoder	77.5	13/31	Data Acquisition Conducted in NTU
Jin et al., (2020) [78]	Bispectrum based channel selection (BCS)	LDA	68.4 76.1 74.9	26/59 24/118 15/60	Dataset I BCI Competition IV Dataset IVa BCI Competition III Dataset IIIa BCI Competition III
Dai et al., (2020) [25]	Shapelet Transform EEG Channel Selection (STCS)	SVM	90.75 77.7 82.63 69.39 77.31 74.4 75.01 70.88 68.55 75.76	4/6 2/7 2/59 16/59 16/59 9/22 11/22 8/22 3/10 7/10	Dataset Ia Dataset Ib Dataset IV 1 calib 1a Dataset IV 1 calib 1b Dataset IV 1 calib 1c Dataset IV 2a s1 Dataset IV 2a s2 Dataset IV 2a s3 Dataset IV 3 s1 Dataset IV 3 s2
Zhang et al., (2021) [32]	Automatic Channel Selection with Sparse Squeeze and Excitation Blocks (ACS-SE)	CNN	87.2	8/60	Motor imagery recognition
Idowu et al., (2021) [22]	Neuro-evolution-ary Algorithm (NEA)	Multilayer Perceptron Neural Network (MPL-NN)	89.95	17/64	EEG Dataset
Shi et al., (2021) [79]	Binary harmony search (BHS)	LDA SVM Sparse Representation classification (SRC)	73.57 77.32 70.35 74.34 78.21 83	17/59 26/118 23/59 31/118 21/59 32/118	Dataset 1 BCI Competition IV Dataset IVa BCI Competition III Dataset 1 BCI Competition IV Dataset IVa BCI Competition III Dataset 1 BCI Competition IV Dataset IVa BCI Competition III
Qi et al., (2021) [80]	Spatio-temporal-Filtering-Based Channel Selection (STECS)	Fisher’s LDA	80.32 90.70 88.49	40/118 25/60 20/32	Dataset IVa BCI Competition III Dataset IIIa BCI Competition III MI experiment provided by Tshingua University
Strypsteen et al., (2021) [24]	Gumbel softmax trick	DNN	91	10/44	High Gamma Dataset

### 6.1. Statistical Analysis

#### 6.1.1. Dataset IVa BCI Competition III

Figure 5 and Figure 6 demonstrate that techniques BSA with CSP, cross-correlation, and mRMR, achieved an average classification accuracy of 94.16%, 99.64%, and 90.77%, respectively, with 24, 71, and 10 of total channels 118. The cross-correlation-based discriminant criterion (XCDS) channel selection technique and the CNN classifier achieved the most remarkable accuracy with 60.16% of channels [21]. However, Table 6 summarizes that the best channel choice technique to create maximum performance for Dataset IVa BCI Competition III had been the mRMR strategy with an average rating of 90.77% with just 8.47% of the 118 correctly classified channels [71]. Figure 5 and Figure 6 demonstrate the classification accuracy and selected channels on dataset IVa BCI competition.

The mRMR approach has the benefit of focusing the typically appropriate channel placements for right hand and foot motor-imagery tasks on the left hemisphere. These illustrate that unilateral MI is concentrated in the contralateral hemisphere’s optimal selection channels. The disadvantage of the mRMR method is that there is no simple positive correlation between increasing the number of channels and improving classification accuracy. In a relatively small number of channels, an increase in channel numbers will significantly improve classification accuracy.

#### 6.1.2. Dataset I BCI Competition IV

This paper has 13 papers that used Dataset I BCI Competition IV and showed promising results using the channel selection technique Bhattacharya bound the sequential forward search (B.B. and SFS) method, and Naïve Bayes (N.B.) classifier. The average classification accuracy was 96.25%, with 50.88% of 59 channels [47,81].

The advantage of the B.B. and SFS method is that it is computed with all possible channels of Bhattacharyya boundaries of CSP functions. The channel set selected by the sequential search strategy is a sub-optimal solution. Figure 7 and Figure 8 demonstrate the classification accuracy and selected channels on dataset IVa BCI competition, respectively. However, it has some limitations, including the need for additional data to provide sure estimate accuracy in a classifier and the difficulty in determining which area of the brain generates class-relevant activity.

#### 6.1.3. Dataset IIIa BCI Competition III

This dataset was utilized in six studies, achieving the best results [63,82]. Jin et al. [59] used a correlation-based CCS channel selection strategy with a classifier SVM and only 31.67% of total channels to reach the metric 91.9%. The advantage of this method is that it is an effective way of choosing channels based on an analysis of the correlation. The CCS algorithm requiring fewer channels can improve the performance of the MI-based BCI and significantly improve the long-term performance of an application with disabled users. The limitation is limited participation and data sizes, leading to an insufficient conviction. Given the context of the correlation analysis, another limitation is the common noise. If there are a lot of common noise components in specific channels, they can be chosen too.

On the other hand, a channel selection technique called spatiotemporal-filtering-based channel selection (STECS) was used with Fisher’s LDA to get 41.67% of total channels [82]. The computational complexity of STECS is modest and can be efficiently implemented to ensure that the algorithm is ideal for online BCI systems in almost real-time. Two drawbacks exist: firstly, STECS should be assessed with additional EEG datasets and superior to other channel selection methods. Second, selecting STECS channels and optimizing multiple spatiotemporal filters are carried out during different phases with different optimizing criteria. Figure 9 and Figure 10 demonstrate the classification accuracy and selected channels on dataset IVa BCI competition.

In the literature, there are numerous comparative studies on channel selection approaches. However, when this research is applied to different datasets, it can produce inconsistent conclusions. Researchers can help to choose the correct channel selection algorithm by examining the current literature under the following sections.

#### 6.1.4. Dataset IIa BCI Competition IV

Figure 11 and Figure 12 demonstrate the classification accuracy and selected channels on dataset IVa BCI competition. With only 45.55% channels, the mRMR approach with SVM obtains 81.22% accuracy, as seen in the figure. The advantage and disadvantage of this method is mentioned in Section 6.1.1.

#### 6.1.5. Other MI Datasets

Figure 13 and Figure 14 show the classification accuracy and selected channels on a different dataset. The trained model weight with the CNN classifier obtained 91.50% accuracy while using 12.50% of channels, as seen in the figure. On other datasets, some approaches reach incredibly high accuracy. Using CNN on (CLA) left/right hand MI datasets, the cross-correlation chooses 78% of the channels and achieves 99.28% accuracy. However, utilizing the trained model weight of CNN, the accuracy is 91.50% compared to 12.50% channels on the dataset of the Chinese Academy of Science. So, using a deep learning model like CNN’s we can select very informative channels and achieve very excellent accuracy.

### 6.2. Machine Learning (ML)/Deep Learning (DL) Based Analysis

We already talked about a detailed analysis in Section 6.1 on various datasets, but we see these in terms of ML/DL-based analysis. In this review, the ML-based classifier, i.e., SVM, and DL-based classifier, i.e., CNN, perform satisfactorily. We can achieve outstanding accuracy using SVM and CNN classifiers with fewer channels. We can see this in Section 6.1 many other datasets have good accuracy but are not promising.

### 6.3. Appropriate Evaluation Criteria

A channel subset must be evaluated for MI applications to establish the best channel subset. Two sorts of measures/criteria may be utilized as assessment criteria: data-based measures/criteria and classification-based measures/criteria.

#### 6.3.1. Information-Based Criteria

Information-based measurements/criteria based on the classification system. The criteria for evaluating a channel subset are needed to determine the optimum channel subset for MI applications. As evaluation criteria, two sorts of measures/criteria can be used. The correlation coefficient, mutual information, symmetrical uncertainty, and others were used to rank channels [83,84]. The distances between the inter-class have been measured with different comparison methods between the binary variables, such as the matching and numerical variables, such as the distance and angular division of the Euclidian [82,85]. A probabilistic dissimilarity measurement was carried out using Chernoff and Bhattacharyya [47,86].

Information-based measurements can be computationally simple, take less time, and require only a single calculation. These metrics are the only viable alternative for applications where time and computational complexity are constraints. The disadvantage is that it does not ensure optimal results.

#### 6.3.2. Classifier-Based Criteria

These measures used classification accuracy to evaluate a channel subset. A classifier oversees determining the separability measure, and when the classifier performs well, a channel is chosen. The classifier’s error rate is a commonly used metric for evaluating the classification method. The disadvantage of classifier assessment is that results will depend on the classifier used and change the performance of the classifier effects. As a result, one can assume that under challenging abilities, the temporal characteristic is more critical than timing [87]. It does, however, ensure the best results. Figure 15 presents the summary of all classifiers used in this paper. The implementation details of the classifier used in this study are mentioned for SVM [1,3,42,46,51,54,56,59,60,62,70,76,79], LDA [26,52,55,58,76,77,78,79], CNN [21,24,27,32], Fisher’s LDA [53,80], FDA [23,43], and other’s classifier [22,25,43,47,57,61,64,73,74,79].

### 6.4. Channel Selection Approaches

Every channel selection method has its own set of features. Approaches independent of the classifier used the statistical properties of channels to filter out weak channels. These solutions rely on the information measure and are independent of the classifier. The use of information measures for channel evaluation makes the filter approach more general and computationally efficient. However, these procedures are typically less effective than other methods. On the other hand, channel selection approaches with a classifier are computer-intensive because a classifier evaluates channel subsets. Therefore, the methodology depends on the classifier but significantly improves classification precision. The advantages and limitations of each method are described in Table 7.

These two characteristics are part of a hybrid channel selection strategy that can be employed to develop flexible methods for channel selection. An innovative technique employs a wide range of channel selection strategies instead of selecting and accepting the results as the final channel subset. Then add the results with an ensemble approach for obtaining the best sub-set channel because several optimal channel sub-sets can exist [88,89]. Figure 16 shows the number of channel selection algorithms used in this paper.

#### 6.4.1. Search Algorithms

The search is complete; the three main algorithms are sequential and heuristic/random search. Each has its own set of pros and cons. Consecutive search is the most frequent search technique, in which channels are added and removed in sequential order. These methods are usually not ideal, but they are simple to build quickly. The B.B. and SFS, GSFS, and SFFS are the most common sequential search methods [47,61,65].

#### 6.4.2. Feature Extraction

Features extraction is also vital in channel selection for EEG applications to maximize the system’s performance. To increase inter-class variability, such feature extraction techniques should be applied. MI applications’ four most used features are time domain, parametric model-based, transformed domain, and frequency-based [90,91]. Features including mean, variance, and the parameter were utilized in the temporal domain, although they were deemed weak features. The auto-regression models and common spatial patterns are model-based characteristics [39,92,93]. The literature also used Fourier and Wavelet to extract features categorized as transformational features. The most frequency domain features in the literature are power spectral density and frequency-based spectral border characteristics [12,94,95,96]. After analyzing the literature on channel selection algorithms, the features recovered by CSP and its variants are more effective.

#### 6.4.3. Feature Selection

Selecting features is also significant for optimizing system performance and choosing appropriate channels. The EEG features collected may have high dimensionality, be redundant, or have outliers that affect the system’s performance. A feature selection algorithm is an option for dealing with these factors. If EEG data are too large, and the calculation cost must also be reduced by using a function selection technique. Using the best functionality to improve classification accuracy or class variance, the best and most appropriate EEG channels are eventually chosen.

### 6.5. Future Direction

Evolutionary algorithms such as neural network-based methods in EEG channel selection are currently being researched and are open to future inquiry. We must thoroughly investigate each technique to create the best channel selection algorithm. We also need to investigate the performance sensitivity of various workloads and classifiers. The automated channel selection method based on deep learning could be a promising future solution for portable EEG-based BCIs. The fewer channels used, the fewer data will be required. As a result, the processing speed and mobility of EEG-based BCIs would be enhanced. In principle, this unique strategy might apply an adaptive-learning approach to diverse BCI paradigms: a technique that could eventually become beneficial for future channel-selection strategies relevant to cutting-edge BCI software platforms, such as Web-based BCIs [97,98]. This study shows that a deep-learning approach to EEG data can yield promising and potent task-relevant EEG characteristics, allowing for practical and ubiquitous channel-selection applications for BCI technology.

## 7. Conclusions

The most comprehensive empirical research of channel selection algorithms for MI classification is presented in this work. The assessment of channel selection algorithms is a complex issue. Different channel selection methods can be evaluated using various factors such as time, complexity, and accuracy. In real-time applications, time and accuracy are the most important criteria. The choice of classifier and subject significantly impacts the performance of channel selection strategies. The classification of motor imageries from selected channels is considerably better with correlation features than with the normalized powers. The high classification may, therefore, offer input for MI training. KMI would be more suited for BCI since its network properties were comparable to motor execution.

The other major issue in selecting channels is the optimum number of channels. The answer to this matter is intricate because the human brain is the most complex entity [99]. It is challenging to generalize EEG decoding methods as even minor experimental modifications can alter signal treatment, feature extraction, and classification procedures. On the other hand, classifier-independent approaches pick the optimum channel subset based on other signal-related criteria. The best channel set depends on application, features, assessment criteria, and channel selection classification. Traditional approaches used criteria to choose the optimum number of channels based, amongst other things, on the convergence of classification accuracy using cross-validation or an analytical solution for an optimization problem. However, the idea of optimum channels is that task information is more conserved than the other channels. Finally, the number of channels, system performance, time, and calculation cost are all factors to consider.

After studying the research, we conclude that the application-relevant brain cortical regions frequently occur in the optimal channel subset. For Dataset IVa BCI Competition III and Dataset IIa BCI competition IV, the mRMR methods are more suitable because the generally applicable channel positions are focused on the left hemisphere and achieve excellent accuracy compared to other techniques. The CNN-based techniques perform well on different datasets with very few channels, apart from BCI competition datasets. For Dataset I BCI Competition IV, the BB and SFS methods are more suitable for the sequential-based method group because they are computed with all possible channels of Bhattacharyya boundaries of CSP functions. Its accuracy on this dataset is very promising, with good amounts of channels compared to other techniques. On the other hand, the correlation-based channel selection strategy shows excellent results for Dataset IIIa BCI Competition III because of its effective method of choosing channels based on correlation analysis.

## Figures and Tables

**Figure 1 bioengineering-09-00726-f001:**
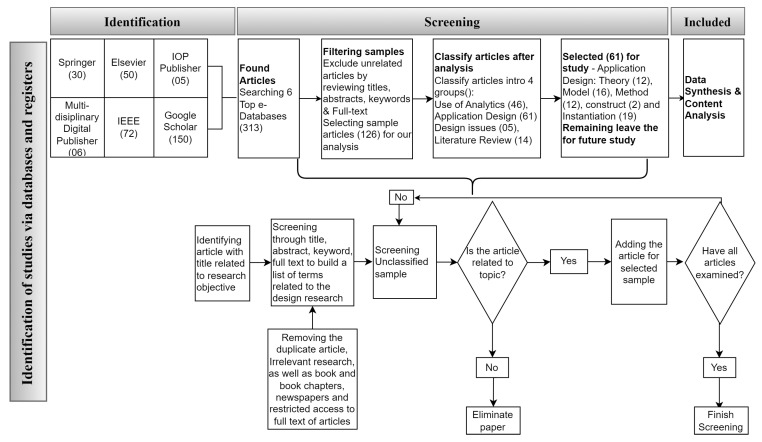
Proposed systematic review for sample collection and analysis.

**Figure 2 bioengineering-09-00726-f002:**
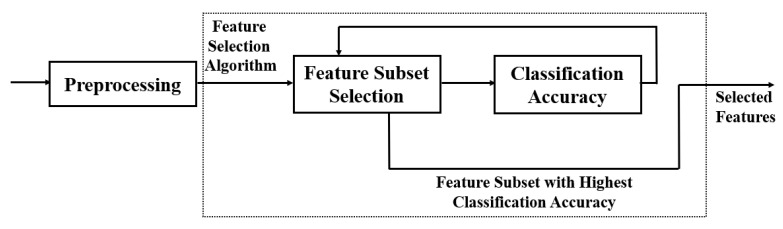
Feature Selection.

**Figure 3 bioengineering-09-00726-f003:**
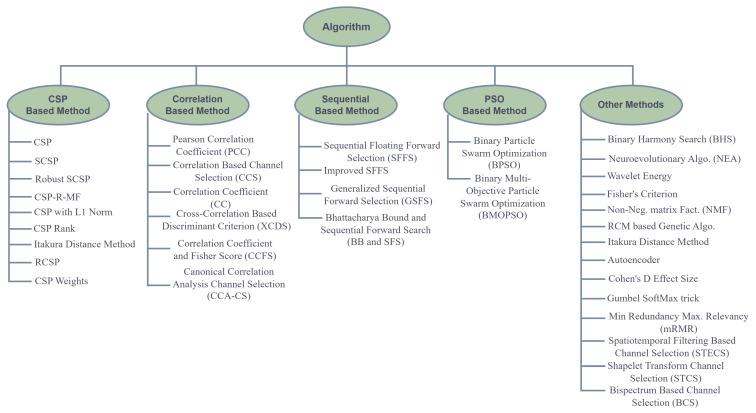
Different EEG channel selection techniques.

**Figure 4 bioengineering-09-00726-f004:**
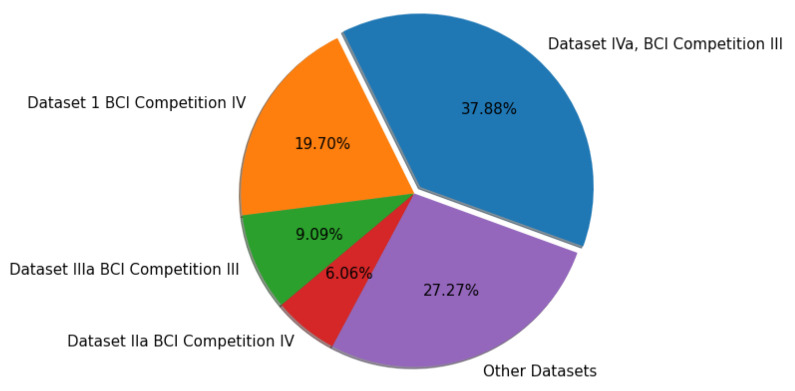
Percentage of different datasets.

**Figure 5 bioengineering-09-00726-f005:**
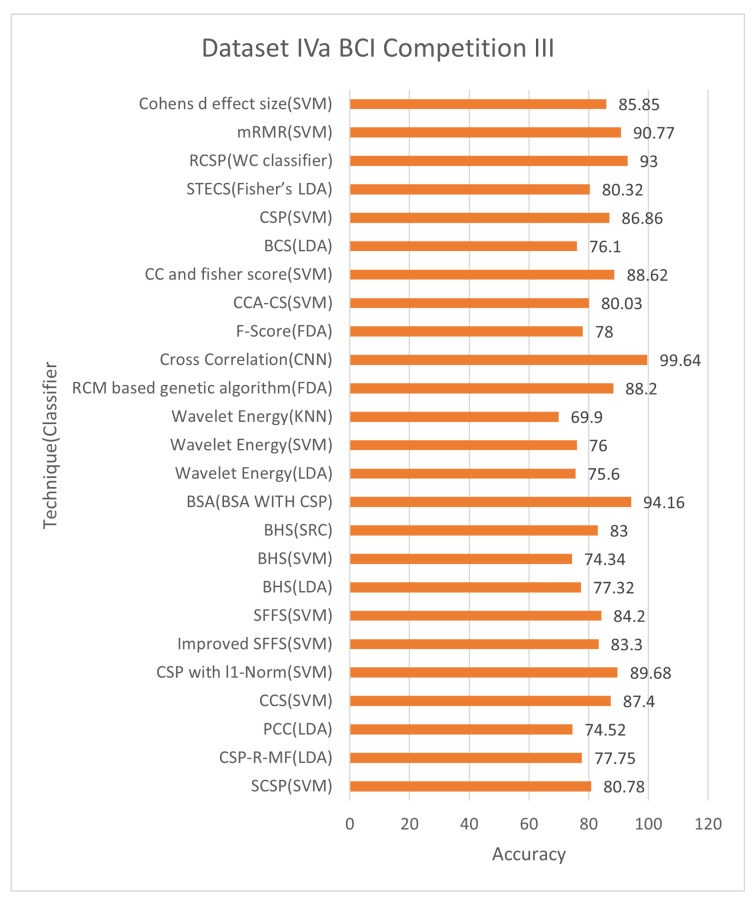
Accuracy by different techniques for Dataset IVa BCI Competition III.

**Figure 6 bioengineering-09-00726-f006:**
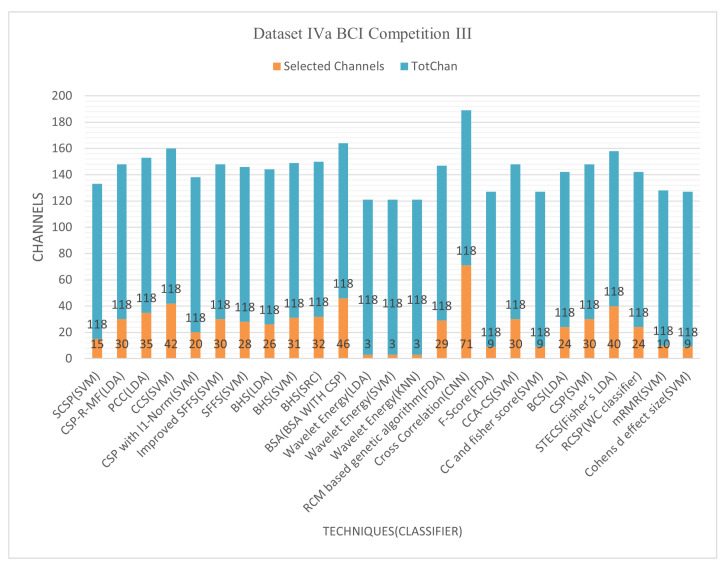
Selected channels by different techniques for Dataset IVa BCI Competition III.

**Figure 7 bioengineering-09-00726-f007:**
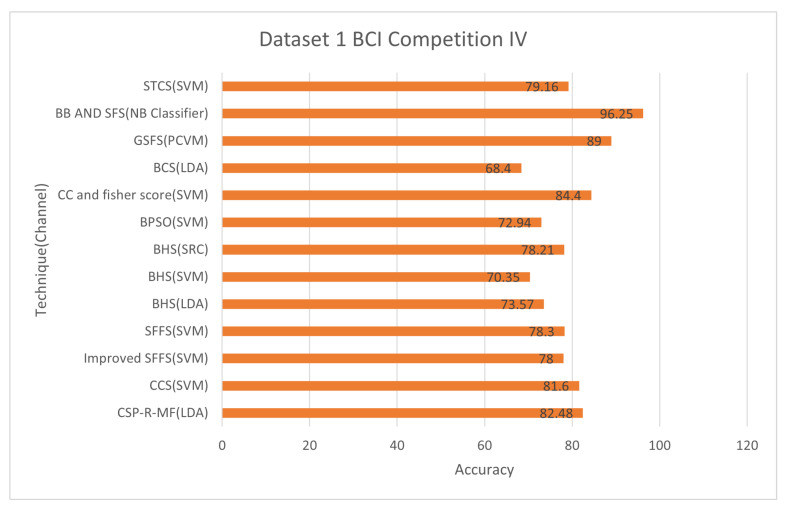
Accuracy by different techniques for Dataset I BCI Competition IV.

**Figure 8 bioengineering-09-00726-f008:**
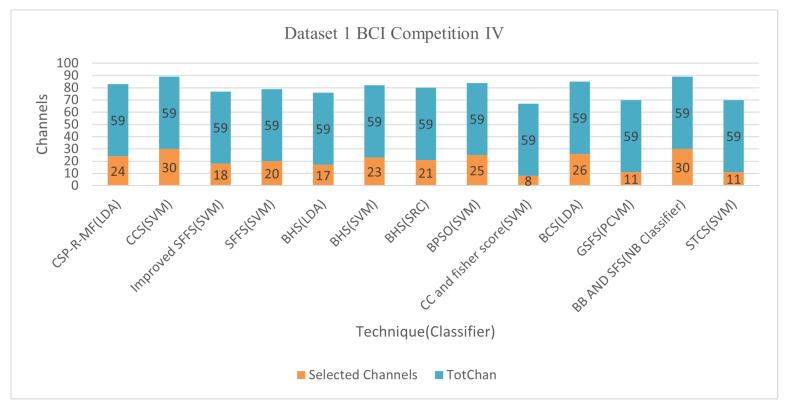
Selected channels by different techniques for Dataset I BCI Competition IV.

**Figure 9 bioengineering-09-00726-f009:**
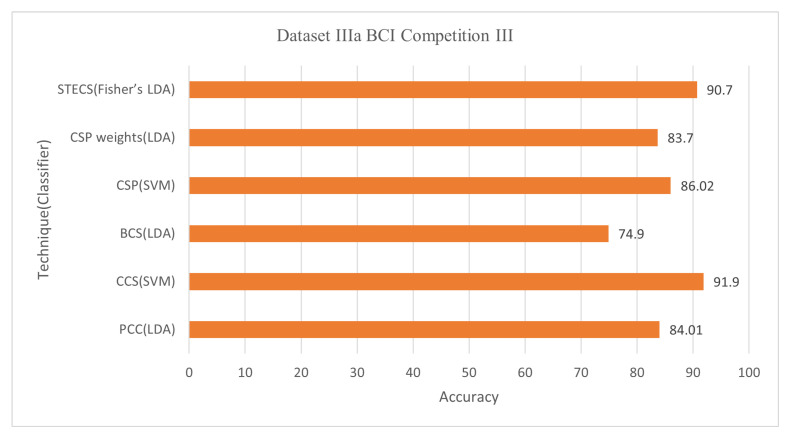
Accuracy by different techniques for Dataset IIIa BCI Competition III.

**Figure 10 bioengineering-09-00726-f010:**
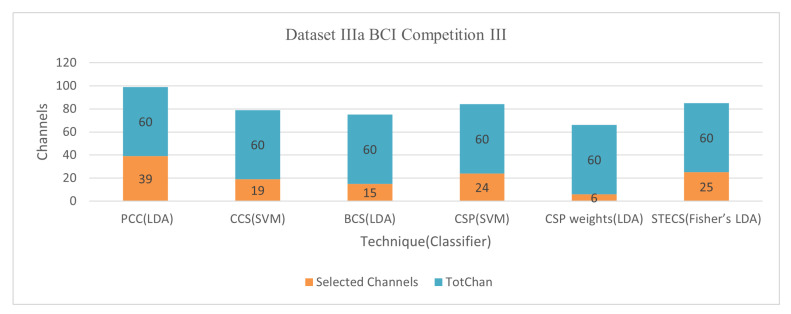
Selected channels by different techniques for Dataset IIIa BCI Competition III.

**Figure 11 bioengineering-09-00726-f011:**
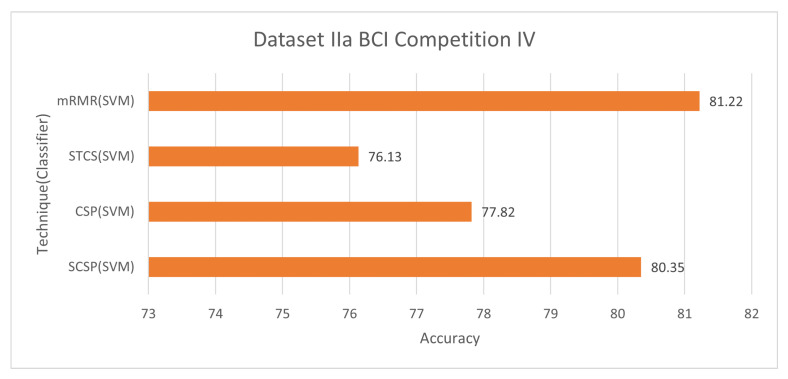
Accuracy by different techniques for Dataset IIa BCI Competition IV.

**Figure 12 bioengineering-09-00726-f012:**
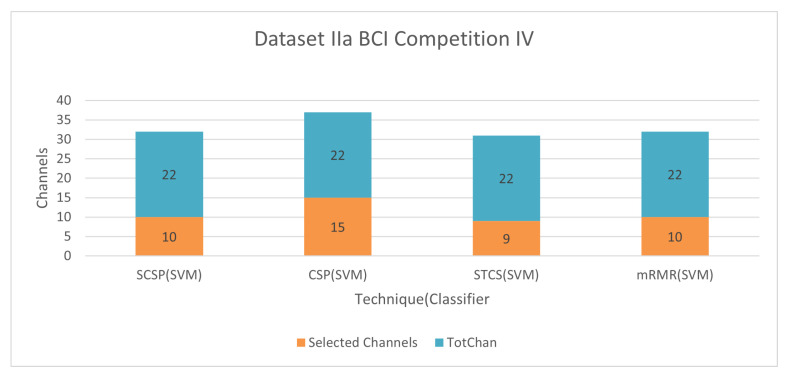
Selected channels by different techniques for Dataset IIa BCI Competition IV.

**Figure 13 bioengineering-09-00726-f013:**
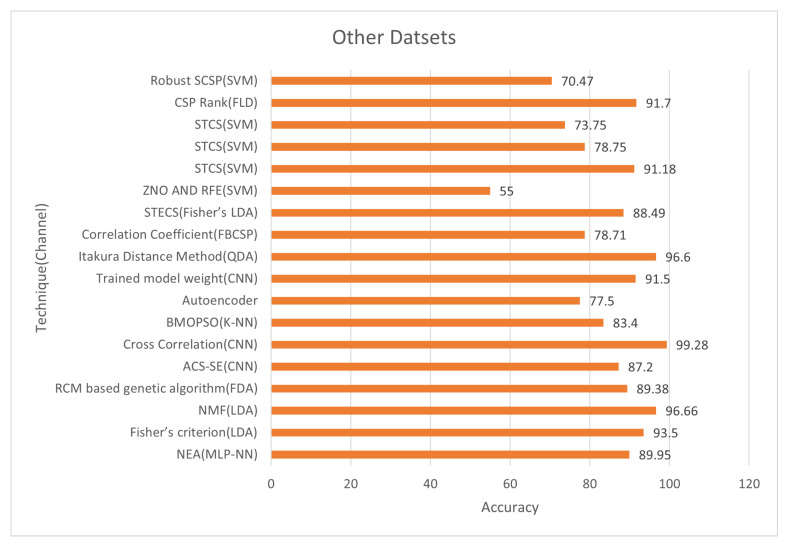
Accuracy by different techniques for different datasets.

**Figure 14 bioengineering-09-00726-f014:**
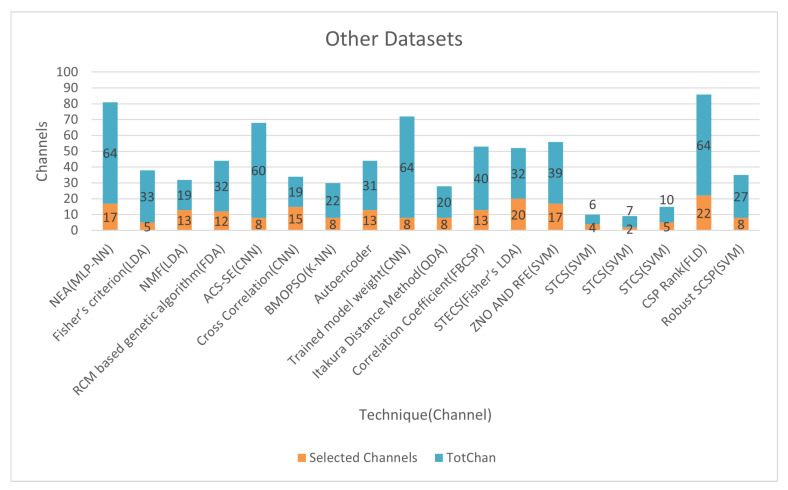
Selected channels by different techniques for different datasets.

**Figure 15 bioengineering-09-00726-f015:**
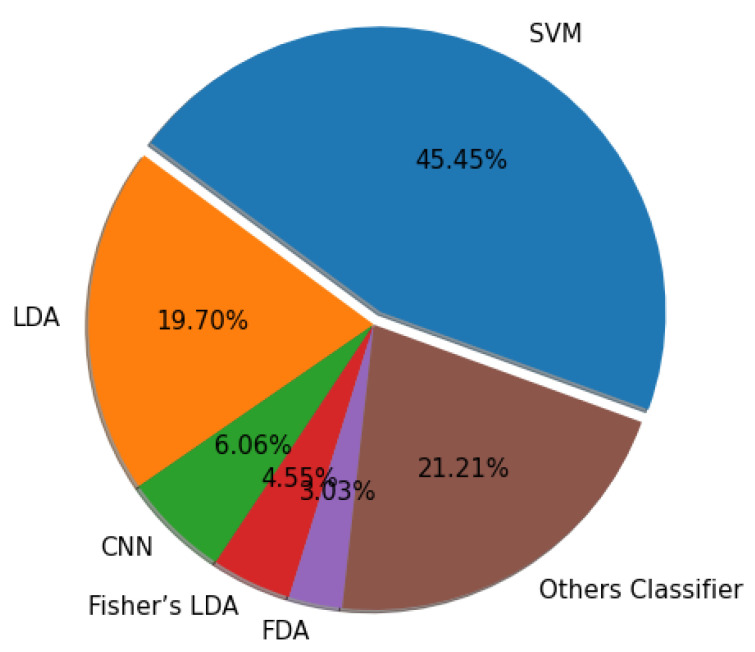
Different classifiers used for channel selections.

**Figure 16 bioengineering-09-00726-f016:**
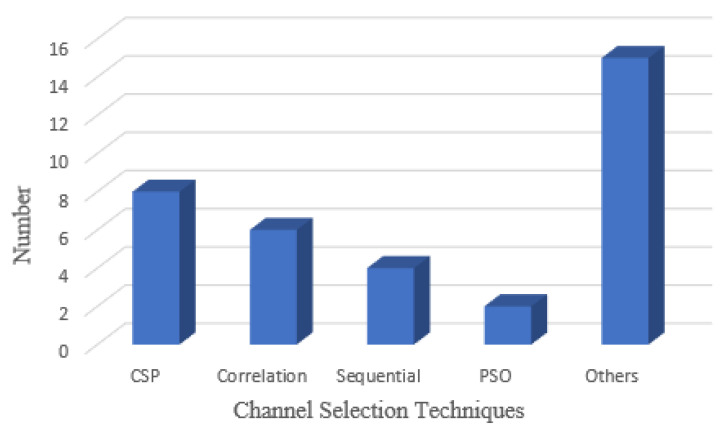
Number of different channels.

**Table 1 bioengineering-09-00726-t001:** Motor Imagery EEG Datasets.

Motor Imagery Datasets	Number of Channels	URL
Dataset IVa BCI Competition III	118	https://www.bbci.de/competition/iii/ (accessed on 27 October 2022)
Dataset 1 BCI Competition IV	59	https://www.bbci.de/competition/iv/ (accessed on 27 October 2022)
Dataset IIIa BCI Competition III	60	https://www.bbci.de/competition/iii/ (accessed on 27 October 2022)
Dataset IIa BCI Competition IV	22	https://www.bbci.de/competition/iv/ (accessed on 27 October 2022)
MI experiments	32	
BCI-FES training platform	64	
Lower limb MI dataset	19	
Dataset Ia BCI Competition II	6	https://www.bbci.de/competition/ii/ (accessed on 27 October 2022)
Amyotrophic Lateral Sclerosis patient	40	
Chinese Academy of Science Data	64	
High Gamma Dataset	44	https://github.com/robintibor/high-gamma-dataset (accessed on 27 October 2022)
Dataset III BCI Competition IV	10	https://www.bbci.de/competition/iv/ (accessed on 27 October 2022)
Data Acquisition was conducted at NTU	31	https://dr.ntu.edu.sg/handle/10356/141527 (accessed on 27 October 2022)
MI recognition	60	
MI Mental Task	39	
MI experiment by Tsinghua University	32	
Stroke patients EEG dataset	27	
Amyotrophic lateral sclerosis (ALS)	33	
(CLA) left/right hand MI dataset	19	https://openbci.com/community/publicly-available-eeg-datasets/ (accessed on 27 October 2022)
G.Tech Gugar Technologies with BCI2000	20	https://www.gtec.at/ (accessed on 27 October 2022)
EEG Dataset	64	
MI Movement	22	
Dataset Ib BCI Competition II	7	https://www.bbci.de/competition/ii/ (accessed on 27 October 2022)

**Table 3 bioengineering-09-00726-t003:** Correlation-based techniques for motor imagery EEG channel selection.

Techniques	Channel Selection Strategy	Classifier	Accuracy (%)	No. of Selected Channels/Total No. of Channels	Dataset
Yang et al., (2018) [43]	Correlation Coefficient	Filter Bank Common Spatial Pattern (FBCSP)	78.71	13/40	Amyotrophic Lateral Sclerosis (ALS) patient
Jin et al., (2019) [59]	Correlation Based Channel Selection	SVM	81.6 87.4 91.9	30/59 42/118 19/60	Dataset 1 BCI Competition IV Dataset IVa BCI Competition III Dataset IIIa BCI Competition III
Park et al., (2020) [60]	Correlation coefficient and fisher score (CCFS)	SVM	88.62 84.4	9/118 8/59	Dataset IVa BCI Competition III Dataset 1 BCI Competition IV
Gaur et al., (2021) [58]	Pearson correlation coefficient (PCC 0.7)	LDA	84.01 74.52	39/60 35/118	Dataset IIIa BCI Competition III Dataset IVa BCI Competition III
Wang et al., (2021) [42]	Canonical correlation Analysis-Channel Selection (CCA-CS)	SVM	80.03	30/118	Dataset IVa BCI Competition III
Yu et al., (2021) [21]	Cross-Correlation based discriminant criterion (XCDS)	Convolutional Neural Network (CNN)	99.64 99.28	71/118 15/19	Dataset IVa BCI Competition III (CLA) left/right hand MI dataset

**Table 4 bioengineering-09-00726-t004:** Sequential-based techniques for the selection of motor imagery EEG channel selection.

Techniques	Channel Selection Strategy	Classifier	Accuracy (%)	No. of Selected Channels/Total No. of Channels	Dataset
He et al., (2009) [47]	Bhattacharyya bound and Sequential forward search (BB and SFS)	Naive Bayes (NB) Classifier	96.25	30/59	Dataset 1 BCI Competition IV
Meng et al., (2011) [62]	Sequential floating forward selection (SFFS)	SVM	78.3 84.2	20/59 28/118	Dataset 1 BCI Competition IV Dataset IVa BCI Competition III
Qiu et al., (2016) [46]	Improved SFFS	SVM	78 83.3	18/59 30/118	Dataset 1 BCI Competition IV Dataset IVa BCI Competition III
Radman et al., (2019) [61]	Generalized Sequential forward selection (GSFS)	Probabilistic classifier vector machine (PCVM)	89	11/59	Dataset 1 BCI Competition IV

**Table 5 bioengineering-09-00726-t005:** Particle swarm optimization-based techniques for motor imagery EEG channel selection.

Techniques	Channel Selection Strategy	Classifier	Accuracy (%)	No. of Selected Channels/Total No. of Channels	Dataset
Kim et al., (2013) [63]	Binary particle swarm optimization (BPSO)	SVM	72.94	25/59	Dataset 1 BCI Competition IV
Wei et al., (2015) [64]	Binary Multi-Objective Particle Swarm Optimization (BMOPSO)	K Nearest Neighbor (K-NN)	83.4	8/22	Motor Imagery (Left hand, Right hand, Foot) Movement

**Table 7 bioengineering-09-00726-t007:** Advantage and limitation of each techniques.

Technique	Advantage	Limitation
CSP	Finds spatial filters for two classes	Noise sensitivity and Overfitting
Correlation based	Most discriminant channels	Performance issue
Sequential based	Achieve better recognition	Slow execution and lack of generality
PSO	Simple to implement and avoiding the unnecessary computations	Slow searching around the global optimum
Others	CNN achieved good results and automatic	Unable to find the best channels

## Data Availability

The data described in this study are accessible from the corresponding author upon request.

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
