# Peer review of "EEG Channel Selection Techniques in Motor Imagery Applications: A Review and New Perspectives"

_bioengineering, 2022, doi:10.3390/bioengineering9120726_

Round 1
Reviewer 1 Report
This paper is a very good, precise, well structured and comprehensive review on EEG channel selection methods in motor imagery applications. The focus of this paper lies on the description and evaluation of the mathematical methods to extract the relevant EEG information. The act of the motor imagery or the graphoelements of the EEG as such are not covered. The mathematical methods are very well described. The inobservance of the underlying biology leads to some odd statements, eg (line 639) that the “The advantage of the mRMR method is that the generally applicable channel positions are focused on the left hemisphere for the right hand”. Of course, movements of the right hand are controlled by the left hemisphere. Apart from that it is a very successful paper.
Minor points:
Line 31 the abbreviation BMI is not well chosen, because everybody knows BMI as body-mass-index
Line 45: medial instead of medical
Line 53 better central brain regions than head regions
Line 226: the authors might specify the used distance norm (euclidian?)
Author Response
"Please see the attachment"

Reviewer 2 Report
This paper reviews the existing works to find the most promising EEG channel selection algorithms, from multi-channel EEG dataset, for effectively identifying the motor imagery tasks. Also the associated classification methodologies on various motor imagery multi-channel EEG datasets are reviewed.
This is a topic of interest and is also a well-researched one. For authors of this paper, I have the following remarks:
1. It is recommended to merge the section “Background” in “Introduction”. For a better presentation it is recommended to split the “Introduction is subsections: 1.1. Background, 1.2. Related works, 1.3. Motivation and contribution.
2. The 4th point of contribution is not clear “We suggest the classifier with the best accuracy for different EEG channel selection techniques”. The “Abstract” gives an impression that authors have reviewed the associated classifiers on various motor imagery multi-channel EEG datasets, therefore the term “suggest” is confusing. Also it is recommended to merge the contributions in 2 points.
3. For the benefit of reader and investigators, it is recommended to add the URLs of datasets discussed in Table 1.
4. Describe the implementation details/hyper-parameters of considered classifiers.
5. Carefully proofread the paper to avoid any typos and grammatical errors.
Author Response
"Please see the attachment"

Reviewer 3 Report
1. It helps to appreciate the paper by having a related review section. The authors should consider more recent research done in the field of their study. Please address the literature systematically. If possible, the authors can give a table pinpointing the advantage or limitations of each work.
2. The article missed presenting the research novelty. In the sense that they do not highlight what is missing from each of the other proposals. The authors should provide enough proof to convince the reader of superiority of the proposed schemes over the existing works.
3. The paper requires further discussion and more details analysis at the manuscript. There is not enough analysis in the review article. Add research directions to the end readers to take this as base paper to conduct more advanced research. The statements should be rephrased/enriched aiming at a clearer exposition and better highlighting the potential research directions.
Author Response
"Please see the attachment"

Round 2
Reviewer 2 Report
Authors have incorporated certain recommended changes in the revised version of manuscript. Howevere, following points still need to be addressed.
1. Please split Introduction in three subsections: 1.1. Context of the study, 1.2. Related works, 1.3. Motivation and contribution.
2. The presented integration of the "Background" section in "Introduction" needs clarification. Therefore, it is recommended to renamed this section as "Basic concepts" and seperate it from the introduction.
3. For the benefit of reader and investigators, it is recommended to add the URLs of datasets (which lead directly to the online available dataset portals) discussed in Table 1.
Reviewer 3 Report
This paper has edited and revised according to the reviewer's suggestions.
